# Preparation of Superhydrophobic Materials and Establishment of Anticorrosive Coatings on the Tinplate Substrate by Alkylation of Graphene Oxide

**DOI:** 10.3390/polym15051280

**Published:** 2023-03-03

**Authors:** Jiangdong Gu, Qiufeng An, Jialong Li, Ping Ge, Yanyan Wu, Yihan Li

**Affiliations:** 1College of Chemistry & Chemical Engineering, Shaanxi University of Science &Technology, Xi’an 710021, China; 2Shaanxi Key Laboratory of Green Preparation and Functionalization for Inorganic Materials, School of Material Science and Engineering, Shaanxi University of Science and Technology, Xi’an 710021, China; 3College of Bioresources Chemical & Materials Engineering, Shaanxi University of Science &Technology, Xi’an 710021, China

**Keywords:** graphene oxide, fluorosilicon polymers, superhydrophobicity, anti-corrosive property

## Abstract

Corrosion of structural parts not only reduces the service life of the equipment but also causes safety accidents, so building a long-lasting anti-corrosion coating on its surface is the key to solving this problem. Under the action of alkali catalysis, n-octyltriethoxysilane (OTES), dimethyldimethoxysilane (DMDMS), and perfluorodecyltrimethoxysilane (FTMS) hydrolyzed and polycondensed co-modified graphene oxide (GO), modified to synthesize a self-cleaning superhydrophobic material fluorosilane-modified graphene oxide (FGO). The structure, film morphology, and properties of FGO were systematically characterized. The results showed that the newly synthesized FGO was successfully modified by long-chain fluorocarbon groups and silanes. FGO presented an uneven and rough morphology on the substrate surface, the water contact angle was 151.3°, and the rolling angle was 3.9°, which caused the coating to exhibit excellent self-cleaning function. Meanwhile, the epoxy polymer/fluorosilane-modified graphene oxide (E-FGO) composite coating adhered to the carbon structural steel’s surface, and its corrosion resistance was detected by the Tafel curve and EIS impedance. It was found that the current density of the 10 wt% E-FGO coating (*I_corr_*) was the lowest (1.087 × 10^−10^ A/cm^2^), which was approximately 3 orders of magnitude lower than that of the unmodified epoxy coating. This was primarily due to the introduction of FGO, which formed a continuous physical barrier in the composite coating and gave the composite coating excellent hydrophobicity. This method might provide new ideas for advances in steel corrosion resistance in the marine sector.

## 1. Introduction

As the most widely used building material, metal corrosion has caused numerous safety disasters and huge economic losses. Therefore, the prevention and mitigation of metal corrosion are of great significance for prolonging the service life of existing structures [1,2,3,4,5,6]. Epoxy resin coatings are widely used to protect metal structures from corrosion due to their strong adhesion, high hardness, good wear resistance, and excellent chemical inertness [7]^.^ However, due to the high temperature of the epoxy resin curing system or the exothermic reaction, the small molecules in the curing system are heated and vaporized, and many holes and defects are formed in the processing process, which leads to the deterioration of its anti-corrosion performance [8,9,10]. In addition, the use of a hydrophilic curing agent leads to a higher moisture absorption rate [11,12,13]. Therefore, determining how to improve the physical barrier and reduce the moisture absorption of epoxy resin coating is the key to solving the corrosion resistance.

The mechanical properties and shrinkage properties of the polymer can be improved by adding nano-filler to the resin [14,15,16,17], for example, SiO_2_, ZnO, TiO_2_, C_3_N_4_, MOF, etc. Ramesh k. et al. [18] used SiO_2_ as a reinforcing filler, and SiO_2_ is embedded in the polymer via the solution intercalation method. The prepared composite coating enhances the adhesion to the metal substrate and fills defects such as cracks and voids inside the polymer film, effectively improving the anti-corrosion ability of the polymer. Graphene is a kind of two-dimensional layered nanomaterial with excellent performance. When the large, layered structure of graphene is used as a nano-filler, it can efficiently inhibit the penetration and penetration of corrosive media into the coating matrix [19]. This can optimize the anti-corrosion resistance of its composite coating [20,21]. However, due to the fact that the graphene sheets are easily superimposed and their poor dispersibility in organic matters, it is difficult to effectively exert excellent properties. Derived from graphene, graphene oxide (GO) possesses a layer structure similar to graphene and good compatibility with organic substances, so it is often used as an anticorrosion filler [22,23]. Guo et al. [24] prepared an aniline trimer-modified GO composite (ATGO) and applied it to improve the anticorrosion property by forming a composite coating with epoxy resin (EP). The data indicate that the corrosion resistance of EP coatings increases by two orders of magnitude after adding 0.05% ATGO. Wang et al. [25] first synthesized γ-(2,3-glycidoxy)propyltrimethoxysilane (GPTS)-functionalized silica nanoparticles (GSiO_2_), then covalently grafted GSiO_2_ onto a green and environmentally friendly nano-hybrid material (GLGO) prepared on lysine-modified graphene oxide (LGO). Finally, GLGO was introduced into the water-based epoxy resin (WEP) coatings. The research illustrated that GLGO could be uniformly distributed in the WEP matrix, and the anti-corrosion performance of the GLGO/WEP coating is better than the WEP coating. This is primarily due to low-surface-energy substances and a unique rough structure at the micro/nanometer scale, which not only traps air but also prevents corrosive media from coming into contact with metal substrates [26]. Du et al. [27] constructed a super hydrophobic composite coating on the surface of aluminum via the combined use of silane/graphene oxide (GO). Through a simple impregnation and curing process, a superhydrophobic silane/graphene oxide (GO) composite coating was systematically synthesized on the aluminum surface. The electrochemical impedance spectroscope (EIS) experiments indicate that the composite coating has the uppermost impedance compared to the exposed Al substrate. The RCT values were 24 and 43 times higher than those of the silane-coated and bare-aluminum substrates, respectively. It can be seen that combining GO with superhydrophobic substances can not only endow it with good hydrophobic function but also dramatically strengthen the anti-corrosion resistance of its composite coating.

In this paper, silane-functionalized graphene oxide (SGO) was synthesized by DMDMS and OTES, and silica was successfully generated on the surface of graphene oxide, which not only can effectively prevent the superposition of GO sheets but also easily construct micro-nano rough structures on its surface. In addition, the hydrophobicity of FGO generated by grafting low-surface-energy fluorosilanes to SGO is significantly improved. By introducing FGO into EP resin, the effect of FGO loading level on the corrosion resistance of EP/FGO (E-FGO) composite coatings was systematically studied. The results showed that the introduction of FGO can effectively improve the anti-corrosion performance of the E-FGO composite coating. This is primarily because FGO can effectively exert physical barrier and hydrophobic characteristics in composite coatings so that composite coatings can achieve the purpose of long-term protection. The experimental results showed that this experiment can provide a new design idea for the modification of anti-corrosion coatings.

## 2. Experimental Section

### 2.1. Materials

Graphene oxide (GO, >99%) was purchased from Kaina Carbon New Materials Co., Ltd., Xiamen, China. Anhydrous ethanol (EA, >97%) and sodium hydroxide (NaOH, Analytical Reagent, >96%) were obtained from Tianjin Hedong District Hongyan Reagent Factory, Tianjin, China. Dimethyldimethoxysilane (DMDMS, Analytical Reagent, 99.7%), perfluorodecyltrimethoxysilane (FTMS, Analytical Reagent, 99.7%), and N-octyl triethoxysilane (OTES, Analytical Reagent, 99.8%) were provided by Tianjin Comeo Chemical Reagent Co., Ltd., Tianjin, China. Epoxy resin (E51, Industrial grade, 0.48~0.54 eq/100 g) was purchased from Changzhou Lebang Composite Materials Co., Ltd., Changzhou, China. Wei Neng ultrasonic cleaner (model CH-02BM, power < 800 W, temperature: room temperature −80 °C, time 0–30 min) was from Suzhou Chuanghui Electronics Co., Ltd., Suzhou, China.

### 2.2. Preparation of Silane-Modified Graphene Oxide (SGO)

Firstly, 100 mL of EA and 50.0 mg of GO were weighed, added to a beaker, and dispersed uniformly and ultrasonically (the power was 300 W ultrasonic dispersion for 30 min at room temperature) to obtain a bright yellow dispersion. Then it was transferred to a 250 mL three-necked flask fitted with a stirrer, a thermometer, and a reflux condenser, and 0.11 g of NaOH, 4.8 g of DMDMS, and 11.06 g of OTES were mixed into the dispersion system, reacted at room temperature for 30 min, and slowly dripped into deionized water for a total amount of 2.3% of the monomers. Next, the reaction was continuously stirred for 24 h in an oil bath (65–75 °C) pot, and the product was filtered with a 0.22 μm polytetrafluoroethylene membrane and repeatedly washed with a large amount of distilled water until neutral. Finally, silane-modified GO (SGO) was obtained via drying.

### 2.3. Preparation of Fluorosilane-Modified Graphene Oxide FGO and E-FGO Composite Coating

We added 60.0 mg of silane-modified graphene oxide (SGO), 120 mL of anhydrous ethanol, and 0.14 g of NaOH into a 250 mL three-necked flask with the assistance of sonication until the additives were uniformly dispersed. We then stirred the mixed solution and stimulated a reaction at 70 °C. Furthermore, 34.0 mg of FTMS and an appropriate amount of EA were added dropwise at 7.2 mL/min to one side of the three-necked flask with a constant-pressure dropping funnel mixture in the system. At the same time, approximately 2.3% of the total monomer was slowly instilled with deionized water at a rate of 7.2 mL/min. After dropping, the reaction continued for 24 h. Ultimately, the product was laundered with distilled water and filtered with a polytetrafluoroethylene (0.22 μm) membrane. Moreover, the product was dried in a vacuum to obtain simple, cheap fluorosilane-modified graphene oxide.

First, we weighed 50 mg of FGO and placed it in the oven to dry and obtain FGO powder. Then we weighed 2.0 g of epoxy resin E51 and diluted it with ethyl acetate solvent to 30% for reserve, added FGO powder and epoxy curing agent D230 to 30% E51 spare resin, and used a high-speed dispersing machine to disperse the mixture at a high speed of 6000 rpm for 15 min. Finally, we put it in the oven to heat up to 120 °C and kept it for 20 min to obtain E-FGO composite coating. The reaction process is shown in Figure 1.

### 2.4. Characterization

The chemical composition of the samples was measured on Fourier transform infrared (FT-IR Bruker vector 22, Bruker-axs) spectra. We observed the microscopic morphology of the sample using a field emission scanning electron microscope (SEM, Nova Nano-SEM, FEI, USA). We recorded the Raman spectrum on a Raman spectrometer (Raman, DXRxi, THEM). X-ray diffraction (XRD, D8ADVANCE-D8X, Bruker, Germany) was applied to perform phase composition analysis using a static contact angle meter to assess the samples of the static water contact angle (CA) and rolling angle (SA). Model DSA20 (Kruss, Germany) of an electrochemical workstation was applied for the anti-corrosion performance, with a three-electrode system in a 3.5 wt% NaCl solution as the electrolyte, where AgCl, Pt, and the test sample function as the reference electrode, the auxiliary electrode, and the working electrode, respectively.

## 3. Results and Discussions

The FT-IR spectra of GO, SGO, and FGO are confirmed in Figure 2a. A wide and strong absorption peak at 3330 cm^−1^ is evident in the FT-IR spectrum of GO, which corresponds to the tensile vibration peak of -OH [28]. The peak at 1726 cm^−1^ is attributed to the C=O stretching vibration peak of the carboxyl group on GO, while the peak at 1618 cm^−1^ should be the bending vibration absorption peak of C-OH [29]. The peaks at 1360–1055 cm^−1^ are the C-O stretching vibration peak and the vibration absorption peak of C-O-C in the carboxyl group, and the two peaks overlap. Apparently, from the FT-IR spectra of SGO and FGO, these four characteristic peaks disappeared or were significantly weakened during the modification process of GO. Moreover, two characteristic absorption peaks at 1040 and 800 cm^−1^ appeared in the FT-IR spectra of SGO and FGO [30]. This should be ascribed to Si-O symmetrical tensile vibration peaks and asymmetric tensile vibrations. The results indicated that -OH, -COOH, and silane on GO were hydrolyzed and condensed during the modification reaction, and thus disappeared. The two small peaks at 2960–2856 cm^−1^ belong to the C-H stretching vibration peaks of -CH_3_ and -CH_2_ in OTES long-chain alkyl groups, and 1276 cm^−1^ shows a large number of C-F bond peaks in FTMS. The low-surface-energy fluorocarbon groups and long-chain alkyl groups are introduced into the structure as explained above. It can be seen that GO was successfully modified into SGO and FGO [31].

The crystal structure of GO, SGO, and FGO was analyzed by XRD. As shown in Figure 2b, the XRD pattern of GO shows sharp and intense peaks at 2θ = 10.56°, which denotes that the GO lattice structure is aligned and highly crystalline. The interlayer spacing of GO can be calculated using the Bragg equation, *d* = 0.837 [32]. However, the XRD pattern of SGO presents slightly wider and weaker dispersion peaks at 2θ = 9.76° and 20.07°, which suggests the structure of SGO becomes a unique dispersion peak of an inorganic–organic hybrid compared with GO [33]. The pattern of FGO showed a broad and strong characteristic dispersion peak around 2θ = 13.29°, and the peak shifted to the higher 2θ angle, indicating that the ordered structure of the modified GO lattice was destroyed. At the same time, owing to a large number of Si-O-Si bonds and functional groups grafted on the GO surface, the interlayer spacing increases, which demonstrates the success of GO modification, which might be advantageous to improve the dispersion of nanofillers [34]. Raman spectroscopy is an important method for characterizing the crystal structure of graphene materials [35,36].

The Raman spectra of GO and SGO and FGO are given in Figure 2c. The width, position, and shape of the Raman spectrum are closely related to the number of layers of GO. It can be seen from the chart that GO has peaks near 1350 cm^−1^ (D peak) and 1580 cm^−1^ (G peak), and the positions of the peaks are particularly close. I_D_/I_G_ is defined as the intensity ratio of the two peaks [37,38], which is an important parameter to characterize the structural defects of graphene-like materials and also measures the disorder degree of the graphene-like material structure [39] The I_D_/I_G_ values of GO, SGO, and FGO were 0.91, 1.04, and 1.05, respectively. GO still retains the original carbon skeleton during the modification process, but the degree of disorder of the structure decreases. This phenomenon is likely due to the grafting of organics such as FTMS onto the GO nanosheets, resulting in increased interplanar spacing and disorder, which is consistent with the XRD results. Meanwhile, two obvious characteristic peaks located at 2916 and 2968 cm^−1^ can be observed in the spectra of SGO and FGO, which is due to the stretching vibration peaks of -CH_3_ and -CH_2_- in long-chain alkyl groups [40]. In conclusion, GO was successfully modified into SGO and FGO while the original 2D structure was retained.

XPS can also be used to detect the chemical composition and surface condition of FGO. Fine spectra of C, Si, and F elements are presented in Figure 2(d1–d3), and the full spectra of FGO, which contains four elements (C, O, Si, and F), are shown in Appendix A. Figure 2(d1) exhibits four peaks at 284.8, 287.1, 291.3, and 293.8 eV of the C1s spectra [25], which can be ascribed to C-Si, O-C=O, C-F_2_, and C-F_3_ bonds [41], respectively. Meanwhile, as shown in Figure 2(d2), the Si2p spectra of FGO can be divided into three peaks at 100.1, 100.9, and 102.2 eV, which correspond to Si-C, Si-O, and O-Si-O bonds, respectively. Figure 2(d3) shows two typical peaks at 689.4 eV and 686.9 eV, which may belong to the C-F_2_ and C-F_3_ binding energies [42], respectively. Overall, the XPS results further reveal a successful combination of FGO consistent with FT-IR, XRD, and Raman results. The SEM images of GO, SGO, and FGO are shown in Appendix A, in which GO shows a sheet-like morphology with obvious edges, and this sheet-like structure provides a larger specific surface area for subsequent alkylation modification. It is noticeable that the surface morphology of the SGO has been significantly changed compared with GO. Meanwhile, the surface of FGO exhibits a large number of micro-nano-sized particle spheres that are evenly distributed, which indicates the roughness of the GO surface is increased.

GO nanosheets present a typical 2D morphology, and the lateral size of GO was approximately 3–5 μm, as shown in Appendix A. TEM images of the FGO are shown in Figure 3a, in which FGO presented a typical silk-like morphology of 2D materials, which demonstrated that the modification process does not destroy the intrinsic structure of GO. Furthermore, the clear folded edges of the nanosheets suggest the thinness of the FGO nanosheets. A high-resolution TEM image of the enlarged area is shown in Figure 3b, in which, in addition to the GO nanosheet, an amorphous layer with a thickness of 5~10 nm can be clearly observed around the nanosheet surface. In addition, the selected area electron diffraction (SAED) of FGO demonstrated in Figure 3b sanctioned the amorphous structure of the shell layer. Importantly, the appearance of F and Si elements in the EDS mapping results further illustrated the successful modification of GO, as shown in Figure 3(c1–c4).

Figure 4a,b show that the water contact angle of the 10% E-FGO coating attained a relatively high value of 151.3° and the rolling angle is as low as 3.9°, which indicates that the 10% E-FGO coating has excellent superhydrophobic properties owing to the hydrophobicity and rough microstructure of FGO presented in Figure 4a. In addition, the droplets of milk, 5% HCl, 5% NaOH, and water spread rapidly when they make contact with the tinplate sheet due to the hydrophilicity of the tinplate surface, as shown in Figure 4c. However, when the counterparts made contact with the tinplate coated with 10% E-FGO, the droplets exhibited a spherical state and did not collapse with the increasing duration, as presented in Figure 4d. Appendix A shows that after the water droplets stayed on the FGO coating for 10 days, the water contact angle was 144.7° and the rolling Angle was 5.3°, which maintained good hydrophobicity. Figure 4(e1–e3) showed when the tinplate surface was covered in contaminants, a large amount of water also failed to carry away pollutants. Figure 4(f1–f3) show that only a small amount of water droplets can dispose of them all on the FGO coating. These results indicated the excellent self-cleaning function of the FGO coating.

## 4. Performance Analysis of E-FGO Composite Coating

Figure 5a,b show the microscopic surfaces of the E-51 and 10% E-FGO composite coatings, from which it can be seen that the surface of the E-51 coating is relatively smooth and flat, and the surface of the 10% E-FGO composite coating has relatively evenly distributed micro-nanoparticles, which effectively increases the surface roughness and improves the hydrophobicity of the E-51 coating. The hydrophobicity increases with the addition of the nano filler (GO, SGO, and FGO), and it was still difficult to make the E-FGO composite layer superhydrophobic when the addition of FGO increased to 15%, which is caused by the hydrophilicity of the E-51 epoxy polymer itself. In general, the FGO superhydrophobic material is well dispersed in the E-51 resin, and the introduction of FGO significantly increases the hydrophobicity of the epoxy polymer, as shown in Figure 5c.

Figure 6a shows the electrochemical impedance and bode spectra of E51, 5% E-FGO, 10% E-FGO, and 15% E-FGO coatings immersed in a 3.5% NaCl solution, where the samples all have a capacitive arc. Moreover, the plots diameter shown in Figure 6a indicates the solution resistance and charge transfer during the corrosion process and is proportional to the corrosion resistance of the sample, where the sample would take on excellent corrosion resistance performance with a large diameter, which is an important parameter for the corrosion resistance performance of the coating [43]. There are obvious differences in the capacitance arc sizes of the four coatings, where the pure E51 coating exhibits the smallest capacitance arc diameter while the 10% E-FGO coating possesses the largest. It can be seen with the doping of FGO that the E51 resin could greatly improve its corrosion resistance. Likewise, when the content of FGO is higher than 10%, the capacitance arc diameter of the E-FGO coating becomes smaller [44].

Figure 6b,c present different bode diagrams of E51 and E-FGO coatings in the low-frequency region (f = 0.01 Hz) tested in a 3.5% NaCl solution, and the impedance modulus values of different coatings change significantly. The induction loop in the low-frequency range of 0.01 Hz–100,000 Hz should be aroused by the relaxation of adsorbed Cl^−^ or H^+^ on the carbon steel surface [45]. When the E-FGO coating concentration increases, the impedance modulus first increases and then decreases, and the 10% concentration showed the highest value, which proves the 10% E-FGO coating possesses the best anti-corrosion effect.

Figure 6d shows the Tafel polarization curve of different samples, with *E_corr_* as the corrosion potential and *I_corr_* as the corrosion current intensity. Clearly, according to the Tafel polarization curve, pure E51 possesses the lowest corrosion potential, which is −0.539 V. However, the corrosion potential of the carbon steel sample coated with E-FGO on the surface of the substrate evidently moves to a higher-position value. When the FGO content is 10%, the corrosion potential of the carbon steel sample coated with E-FGO reached the highest level of −0.487 V. Evidently, the introduction of FGO can provide an effective barrier between the substrate and the environment and has a good protective function [46,47,48]. Moreover, the *I_corr_* of 10% E-FGO is the lowest (1.087 × 10^−10^ A/cm^2^), at only one-thousandth of the other samples. The low *I_corr_* of 10% E-FGO is predominantly due to the excellent barrier effect of FGO, which can effectively prevent the corrosive medium from entering the substrate. However, as the FGO content exceeds 10%, its *I_corr_* becomes larger. This is primarily ascribed to the excess introduction of FGO leading to its aggregation in the composite coating, thereby adding more defects and accelerating the entry of corrosive media [49,50].

Figure 7 shows the equivalent circuit diagram of E51 and E-FGO, where Rs represents the solution resistance and Ccoat is the constant phase element for preparing the coating. Rcoat represents the resistance of the coating during preparation, Rct is the transfer charge resistance, and Cdl is the constant phase element of the electric double layer.

Table 1 lists the electrical parameters of the E-51 coating and the F-GO coating in a 3.5 wt% NaCl solution. The E-FGO composite coating has lower capacitance (Cdl) and higher resistance (Rct) than the E51 coating, indicating that the addition of FGO improves the corrosion resistance of the E51 coating.

The accurate corrosion resistance properties of the as-synthesized sample have been evaluated by salt-spray resistance, acid, and alkali resistance tests. Figure 8 illustrated the salt-spray resistance of different samples. Clearly, compared with a large number of corrosion spots on the pure epoxy sample, the corrosion resistance of the 10% E-FGO-coated sample shows a large increase with fewer corrosion spots, as well as reduced scratch corrosion expansion. The acid resistance test of the pure epoxy resin indicates that multi-point corrosion occurred on the coatings, while the alkali resistance experiment of the pure epoxy coating shows that foaming and shedding ooze occurs on the surface. Conversely, the 10% E-FGO coating presents the opposite results in both acid/alkali resistance experiments. These indicate that the introduction of FGO can considerably optimize the salt, acid, and alkaline resistance of the epoxy coating. Upon further increasing the E-FGO concentration to 15%, the corrosion resistance of the composite coating inevitably decreases. This is predominantly due to the fact that FGO has a sizeable relevant surface area, and when abundant FGO is incorporated into epoxy coatings, agglomeration tends to occur. A small amount of FGO can exert a hydrophobic effect, shielding impact, and labyrinth effect in the coating, thus significantly heightening the anti-corrosion performance of epoxy coatings. Finally, a summary table comparing this work to related studies is shown in Appendix A. The hydrophobicity of the modified GO in this paper was 151.3°, and the anti-corrosion performance of the composite coating was three orders of magnitude higher than that of the pure epoxy coating, and the anti-corrosion performance is excellent.

## 5. Conclusions

In summary, fluorine-containing silane-modified graphene oxide (FGO) with superhydrophobic and self-cleaning functions was successfully synthesized. FGO has superhydrophobicity and excellent self-cleaning functions due to low-surface-energy fluorine-containing substances and the rough surface micro-nano structure. The FGO presents superhydrophobicity and self-cleaning functions evidenced by the contact angle (151.3°) of the FGO coating and the rolling angle (3.9°). Ultimately, E51 resin was initiated to study the anti-corrosion properties of its composite coating with extremely low *I_corr_* of 1.087 × 10^−10^ A/cm^2^, and a significant reduction of 3 orders of magnitude over the neat E51 coating was achieved in the E-FGO composite coating with the loading of 10% FGO.

## Figures and Tables

**Figure 1 polymers-15-01280-f001:**
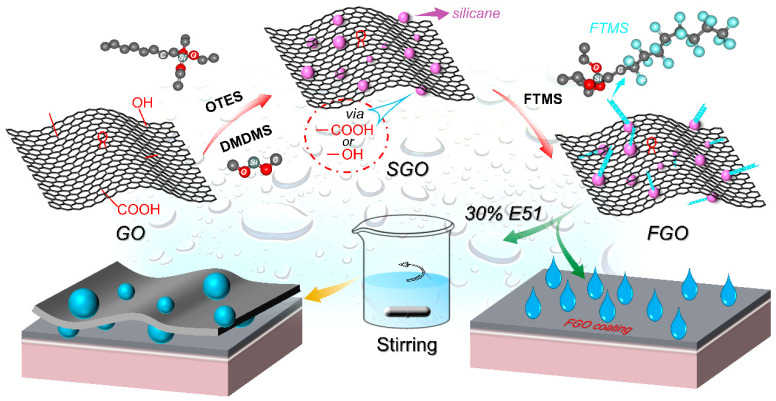
Diagram of preparation of FGO and composite coating E-FGO.

**Figure 2 polymers-15-01280-f002:**
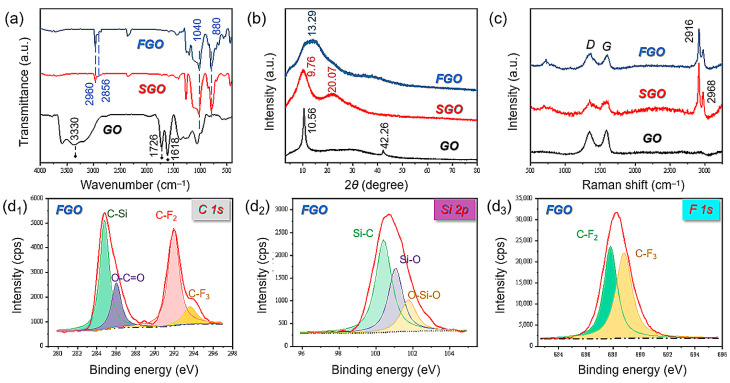
(**a**) FT-IR, (**b**) XRD, (**c**) Raman, and (**d1**–**d3**) XPS spectra patterns of GO, SGO, and FGO.

**Figure 3 polymers-15-01280-f003:**
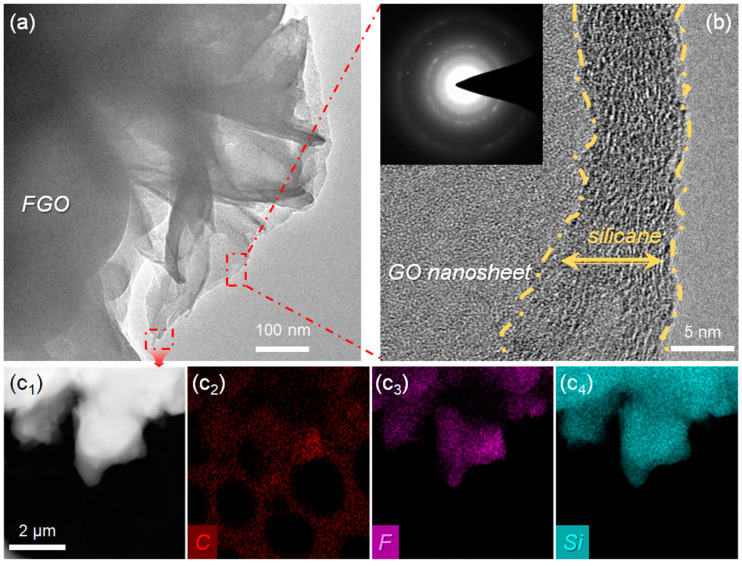
(**a**) TEM image, (**b**) high-resolution TEM image (the inset is the selected area electron diffraction), and (**c1**–**c4**) EDS mapping of FGO.

**Figure 4 polymers-15-01280-f004:**
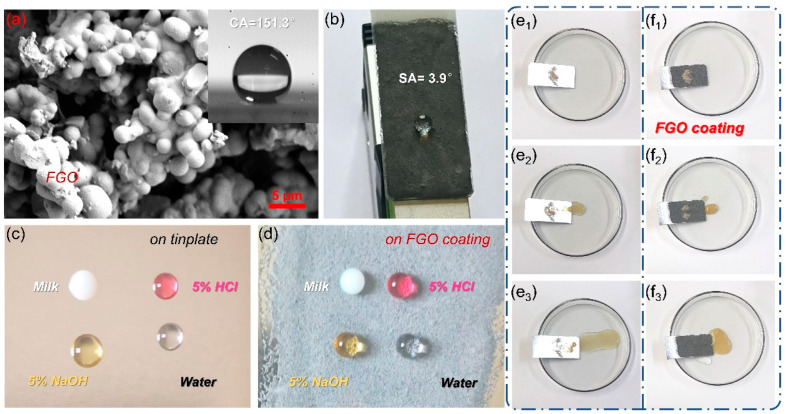
(**a**) State diagram of rolling angle and (**b**) water contact angle of FGO coating. Performance diagram of different liquid drops on the (**c**) tinplate and (**d**) FGO coating and self-cleaning diagram of the (**e1**–**e3**) tinplate and (**f1**–**f3**) FGO coating.

**Figure 5 polymers-15-01280-f005:**
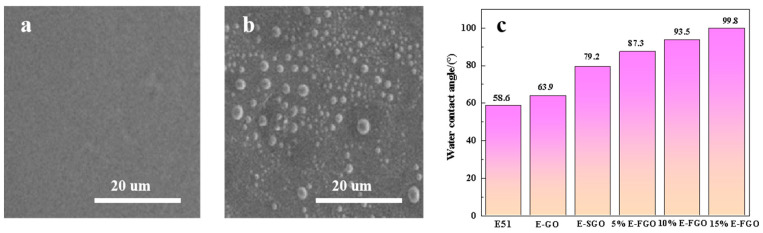
(**a**,**b**) FESEM plots of E-51 and 10% E-FGO composite tinting layers, respectively, and (**c**) the effect of different proportions of (GO, SGO, and FGO) on E-51 water contact angle.

**Figure 6 polymers-15-01280-f006:**
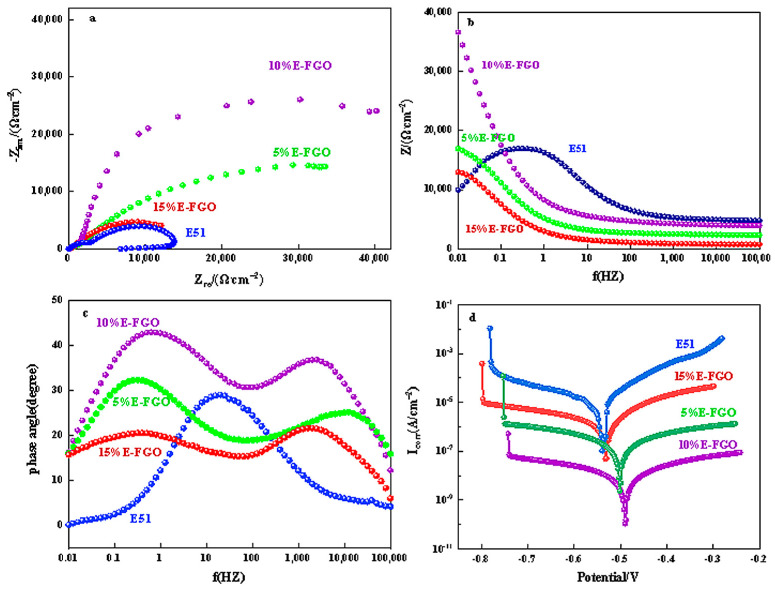
(**a**) Nyquist, (**b**,**c**) Bode plots and phase angle, (**d**) and Tafel curves of E51 and E-FGO coatings.

**Figure 7 polymers-15-01280-f007:**
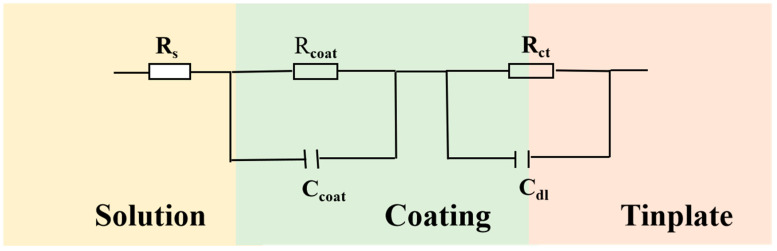
Equivalent circuits of E51 and E-FGO coatings.

**Figure 8 polymers-15-01280-f008:**
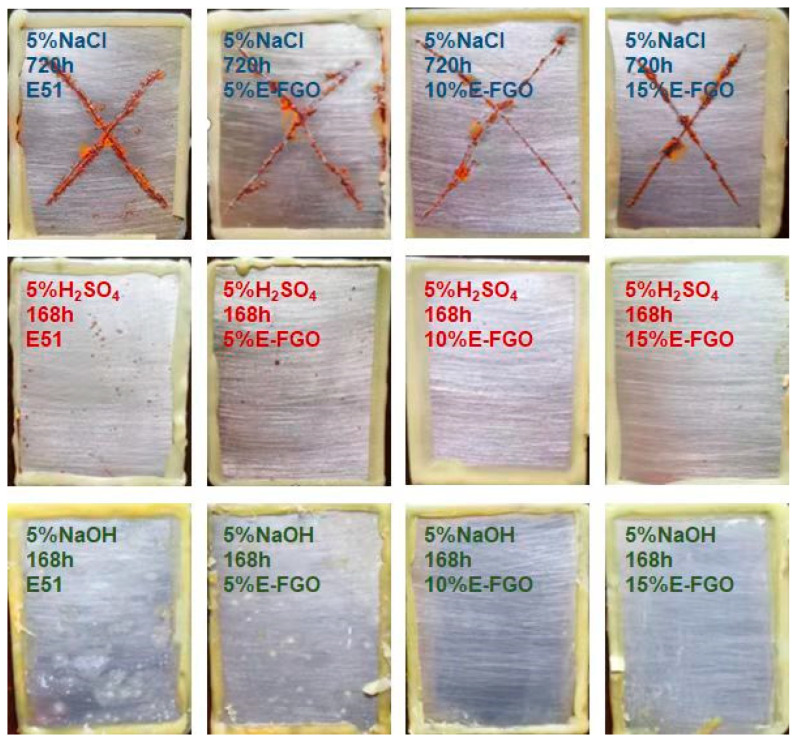
Salt-spray resistance, acid, and alkali resistance tests of E51 and different concentrations of E-FGO coatings.

**Table 1 polymers-15-01280-t001:** Electrochemical parameters for E51 and E-FGO coatings in 3.5 wt% NaCl solution.

Sample	CPE_c_	R_c_	CPE_dl_	R_ct_
E51	7.42 × 10^−4^	1.10 × 10^4^	1.90 × 10^−4^	15.91
5% E-FGO	5.05 × 10^−5^	6.83 × 10^4^	3.42 × 10^−5^	25.85
10% E-FGO	2.61 × 10^−5^	8.27 × 10^4^	2.99 × 10^−5^	169.10
15% E-FGO	1.22 × 10^−4^	1.89 × 10^4^	2.73 × 10^−5^	23.08

## Data Availability

The data presented in this study are available on request from the corresponding author.

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
