# Peer review of "Preparation of Superhydrophobic Materials and Establishment of Anticorrosive Coatings on the Tinplate Substrate by Alkylation of Graphene Oxide"

_polymers, 2023, doi:10.3390/polym15051280_

Round 1

Reviewer 1 Report

Overall, the manuscript is organized and written professionally. Minor revision is required before the paper can be accepted for publication. The detailed comments are as follows.

Abstract – A problem statement should be provided in the beginning of abstract. Besides, please correct the typo for word “which” (Line 20).

Introduction – Authors reviewed several studies related to GO for anticorrosion enhancement, but the problem statement is not clear. What are the novelty of the current work and what is the key knowledge gap to be addressed by this study?

Line 181-184 – Please indicate the figure number properly. It should be “Figure 2d1”, not Figure d2!

Figure 3 – I strongly advise the authors to include TEM/HRTEM of GO sample as well.

Figure 4e1-e3 and f1-f3 – The discussion on these images is not clear. I don’t know what are the findings the authors would like to share. Besides, I advise the authors to include reusability of the FGO coated sample for multiple testing durations.

A summary table comparing the best findings of this work with relevant studies is recommended.  

Lastly, please discuss the cost of fabrication using FGO and the practicability of FGO for industrial adoption.

Author Response

Re: Revisions requested of Manuscript ID: polymers-2151021.

Thank you very much for giving us an opportunity to revise the manuscript entitled “Preparation of superhydrophobic materials and construction of anticorrosive coatings on the tinplate substrate by alkylation of graphene oxide”. Your comments and those of the reviewers are highly insightful and enable us to greatly improve the quality of our manuscript. In the following pages are our point-by-point responses to each of the comments of you. Revisions in the manuscript are shown using yellow highlight [example] for changes.

Referee: 1

Comments to the Author

Overall, the manuscript is organized and written professionally. Minor revision is required before the paper can be accepted for publication. The detailed comments are as follows.

1)Abstract – A problem statement should be provided in the beginning of abstract. Besides, please correct the typo for word “which” (Line 20).

Answer: Many thanks for your attention to our work. In accordance with this suggestion,We have provided the problem statement in the beginning of abstract, and “which” have corrected in the manuscript.

Corrosion of structural parts not only reduced the service life of the equipment but also easily caused safety accidents, so building a long-lasting anti-corrosion coating on its surface is the key to solving the problem. And line 20 has been corrected.

and the rolling angle was 3.9°, which made the coating exhibit excellent self-cleaning function.

2)Introduction – Authors reviewed several studies related to GO for anticorrosion enhancement, but the problem statement is not clear. What is the novelty of the current work and what is the key knowledge gap to be addressed by this study?

Answer: Many thanks for your attention to our work. We have added the novelty of current work and key knowledge gap at the end of introduction.

In this paper, silane-functionalized graphene oxide SGO was synthesized by DMDMS and OTES, and silica was successfully generated on the surface of graphene oxide, which can not only effectively prevent the superposition of GO sheets, but also easily construct micro-nano rough structures on its surface. In addition, the hydrophobicity of FGO generated by grafting low surface energy fluorosilanes to SGO is significantly improved. By introducing FGO into EP resin, the effect of FGO loading level on the corrosion resistance of EP/FGO (E-FGO) composite coatings was systematically studied. The results showed that the introduction of FGO can effectively improve the anti-corrosion performance of E-FGO composite coating. This is mainly because FGO can effectively exert physical barrier and hydrophobic characteristics in composite coatings, so that composite coatings can achieve the purpose of long-term protection. The experimental results showed that this experiment can provide a new design idea for the modification of anti-corrosion coatings.

3)Line 181-184 – Please indicate the figure number properly. It should be “Figure 2d1”, not Figure d2!

Answer: Many thanks for your attention to our work. The figure number has been revised in the manuscript.

4)Figure 3 – I strongly advise the authors to include TEM/HRTEM of GO sample as well.

Answer: Many thanks for your attention to our work. The sample GO of TEM/HRTEM have put in the supporting materials.

Fig.S3 TEM of GO.

GO nanosheets present typical 2D morphology,  the lateral size of GO is about 3-5 um, as shown in Fig. S3.

5)Figure 4e1-e3 and f1-f3 – The discussion on these images is not clear. I don’t know what are the findings the authors would like to share. Besides, I advise the authors to include reusability of the FGO coated sample for multiple testing durations.

Answer: Many thanks for your attention to our work. According to your suggestion, we have discussed figure 4e1-e3 and f1-f3. Meanwhile, we have included reusability of the FGO coated sample for multiple testing durations, all of them have put in supporting-file (Table S1).

Fig. 4e1~e3 showed when the tinplate surface was covered with contaminants, that a large amount of water also falt to carry away pollutants. Fig. 4f1~f3 shows only a small amount of water droplets can take them all away on the FGO coating. These results manifested the excellent self-cleaning function of the FGO coating.

Table S1. Effect of water droplet residence time on FGO coating on water contact angle and rolling angle

T(d)

CA (°)

SA (°)

1

151.3°

3.9°

3

150.9°

4.2°

5

150.1°

4.8°

10

144.7°

5.3°

Note: Time in days (T), superhydrophobic water contact angle (CA), superhydrophobic water rolling angle (SA).

Table S1 shows that after the water droplets stay on the FGO coating for 10 days, the water contact angle is 144° and the rolling Angle is 5.3°, which is still maintain good hydrophobic.

6)A summary table comparing the best findings of this work with relevant studies is recommended.

Answer: Many thanks for your attention to our work. Base on your recommendation we have made a summary table S2 in supporting materials to compare with relevant studies.

Table S2 This work and related literature hydrophobic properties and corrosion resistance comparison table

Sample

CA

Nrelative Icorr

Reference

ATGO

_

2

[22]

GLGO

_

1

[23]

Silane/GO

< 151°

-

[25]

SF-GO

_

4

[51]

FGO

151.3°

3

Our work

The difference between the current density of the composite coating and the current density of the unmodified epoxy coating(Nrelative Icorr), Superhydrophobic water contact angle (CA), The non-superhydrophobic and unmeasured composite coatings (—).

In the end, a summary table comparing this work to related studies have shown in Table S2. The hydrophobicity of the modified GO in this paper was 151.3°, and the anti-corrosion performance of the composite coating was three orders of magnitude higher than that of the pure epoxy coating, and the anti-corrosion performance is excellent.

7)Lastly, please discuss the cost of fabrication using FGO and the practicability of FGO for industrial adoption.

Answer: Many thanks for your attention to our work. We have calculated the practical cost of FGO

The cost is GO ¥50 /g OTES ¥2/g DMDMS ¥2.2/g FTMS ¥14.8/g, which is a good way to obtain simple and cheap fluorosilane-modified graphene oxide. According to the synthesis method in the text, it takes about ¥33 to calculate 1g FGO Commercial application: FGO can improve the comprehensive properties of coatings such as anti-corrosion, anti-fouling, flame retardant and wear-resistant in commerce.

And the product was dried in vacuum to obtain simple and cheap fluorosilane-modified graphene oxide.

Reviewer 2 Report

I have read the manuscript provided by the authors, and I find it quite interesting since polymeric coatings are quite an important field. I have some questions and some points that need to be clarified. Please refer to the pdf attached.

Author Response

Re: Revisions requested of Manuscript ID: polymers-2151021.

Thank you very much for giving us an opportunity to revise the manuscript entitled “Preparation of superhydrophobic materials and construction of anticorrosive coatings on the tinplate substratel by alkylation of graphene oxide”. Your comments and those of the reviewers are highly insightful and enable us to greatly improve the quality of our manuscript. In the following pages are our point-by-point responses to each of the comments of you. Revisions in the manuscript are shown using yellow highlight [example] for changes.

1) Title: I suggest changing the word “construction” with a more appropriate.

Answer: Many thanks for your attention to our work. The word “construction” in the title has been replaced by the word “establishment”.

Preparation of superhydrophobic materials and establishment of anticorrosive coatings on the tinplate substrate by alkylation of graphene oxide

2)Lines 43-44: I think authors should refer which are these nanofillers and elaborate a little bit more this point.

Answer: We have referred to the nanofillers and detailed them in the manuscript.

The mechanical properties and shrinkage properties of the polymer can be improved by adding nano-filler to the resin [14-17] For example, SiO2, ZnO, TiO2, C3N4, MOF, and so on. Ramesh k. et al [18] use SiO2 as a reinforcing filler, and SiO2 is embedded in the polymer by solution intercalation method. The prepared composite coating enhances the adhesion to the metal substrate and fills the defects such as cracks and voids inside the polymer film, effectively improving the anti-corrosion ability of the polymer. And relevant details have been added to the manuscript.

[18] Ammar. S, Ramesh. K, Vengadaesvaran. B, Ramesh. S, Arof. A. K. A novel coating material that uses nano-sized SiO2 particles to intensify hydrophobicity and corrosion protection properties. Electrochim. Acta., 2016, 220, 417-426.

3)Lines 48-50: Please add references

Answer: The reference [19] has been added in lines 48-50

[19] Ding. J. H, Zhao. H. R, Ji. D, Xu. B. Y, Zhao. X. P, Wang. Z, Wang. D L, Zhou. Q. B, Yu. H. B.Achieving long-term anticorrosion via the inhibition of graphene’s electrical Activity, J. Mater. Chem., 2019, 7(6): 2864-2874.

4)Introduction: I think authors should add some works on the use of other fillers in

preparation of anticorrosive coatings.

Answer: Many thanks for your attention to our work. We have elaborated our work in preparation of anticorrosive coatings according to the reviewer’s suggestion.

In this paper, silane-functionalized graphene oxide SGO was synthesized by DMDMS and OTES, and silica was successfully generated on the surface of graphene oxide, which can not only effectively prevent the superposition of GO sheets, but also easily construct micro-nano rough structures on its surface. In addition, the hydrophobicity of FGO generated by grafting low surface energy fluorosilanes to SGO is significantly improved. By introducing FGO into EP resin, the effect of FGO loading level on the corrosion resistance of EP/FGO (E-FGO) composite coatings was systematically studied. The results showed that the introduction of FGO can effectively improve the anti-corrosion performance of E-FGO composite coating. This is mainly because FGO can effectively exert physical barrier and hydrophobic characteristics in composite coatings, so that composite coatings can achieve the purpose of long-term protection. The experimental results showed that this experiment can provide a new design idea for the modification of anti-corrosion coatings.

5)2.1: Please add purities of the materials used.

Answer: Many thanks for your attention to our work. The purities of the materials used in our study have added in the manuscript.

Graphene oxide (GO, >99%) was purchased from Kaina Carbon New Materials Co., Ltd. Anhydrous ethanol (EA, >97%) and sodium hydroxide (NaOH, Analytical Reagent,>96%) were obtained from Tianjin Hedong District Hongyan Reagent Factory. Dimethyldimethoxysilane (DMDMS, Analytical Reagent, 99.7%), perfluorodecyltrimethoxysilane (FTMS, Analytical Reagent, 99.7%) and N-octyl triethoxysilane (OTES, Analytical Reagent, 99.8%) were provided by Tianjin Comeo Chemical Reagent Co., Ltd. Epoxy resin (E51, Industrial grade, 0.48~0.54 eq/100g) , was purchased from Changzhou Lebang Composite Materials Co., Ltd.

6)Line 94: Indicate more information on the sonication. Temperature, sonication tip or bath, cycle-power and time.

Answer: Many thanks for your attention to our work. It has been added to the material reagent part.

Wei Neng ultrasonic cleaner (model CH-02BM, power< 800W, temperature: room temperature -80 °C, time 0-30 min), Suzhou Chuanghui Electronics Co., Ltd

7)2.2: How much is the yield of this reaction? Is this 100%?

Answer: Many thanks for your attention to our work. The yield of this reaction is >90%,

8)Line 104: What is absolute ethanol? Before, authors used anhydrous.

Answer: Many thanks for your attention to our work. The absolute ethanol and the anhydrous are the same substance. It is an error we made in the manuscript, which have been harmonized and revised.

9)Line 105: Again, define the sonication parameters.

Answer: Many thanks for your attention to our work. The sonication parameters have been defined.

The power was 300W ultrasonic dispersion for 30 minutes at room temperature.

10)Figure 1: Authors refer in the text that this is the reaction modification process of the silanized graphene. Instead, the figure shows the whole reaction from the initial graphene and the application as coating. I think authors should change the figure or add more information in figure caption and inside the text.

Answer: Thank you for your suggestion. The corresponding caption has been added to the corresponding image and stated in the text section.

2.3. Preparation of fluorosilane-modified graphene oxide FGO and E-FGO composite coating

Adding 60.0 mg silane-modified graphene oxide (SGO), 120 mL anhydrous ethanol and 0.14 g NaOH into a 250 mL three-necked flask with the assistance of sonication until the additives were uniformly dispersed. Then stir the mixed solution and reacted at 70°C.and 34.0 mg of FTMS and an appropriate amount of EA were added dropwise at 7.2 mL/min to one side of the three-necked flask with a constant pressure dropping funnel mixture into the system. At the same time, about 2.3% of the total monomer is slowly instilled with deionized water at a rate of 7.2 mL/min. After dropping, the reaction continues for 24 h. Ultimately, the product was laundered with distilled water and filtered  with a polytetrafluoroethylene (0.22 μm) membrane. And the product was dried in vacuum to obtain fluorosilane-modified graphene oxide.

First weigh 50 mg of FGO into the oven to dry and obtain FGO powder, then weigh 2.0 g of epoxy resin E51 and dilute it with ethyl acetate solvent to 30% for reserve, and then add FGO powder and epoxy curing agent D230 to 30% E51 spare resin, use a high-speed dispersing machine to disperse the mixture at high speed at 6000 rpm for 15 min. Finally, put it into the oven to heat up to 120 °C and keep it for 20 min to obtain E-FGO composite coating. The reaction process is shown in Fig.1.

Figure 1. Diagram of preparation of FGO and E-FGO composite coating.

11)2.3: What is the yield of the reaction? Why here no washing were done and only

filtered?

Answer: Many thanks for your attention to our work. The yield is 92.2%, and the text says filter and laundered. This has been corrected in the text.

 Ultimately, the product was laundered with distilled water and filtered with a polytetrafluoroethylene (0.22 μm) membrane.

12)Figure 2: This figure needs improvement in quality, please provide a better quality

image. The numbers seem blurry.

Answer: Many thanks for your attention to our work. We have replaced figure 2 to a clear image.

(The current sharpness is 1000dpi, the cause of the blur may be the phenomenon caused by PDF compression.)

Figure 2. (a) FT-IR, (b) XRD, (c) Raman and (d) XPS spectra patterns of GO, SGO and FGO.

13)FTIR, RAMAN, XRD and XPS are all in the same text being a little confusing. I suggest to the author to have subsections with each technique.

Answer:Thank you for your suggestion. The process of expression, FTIR, RAMAN, XRD and XPS have been divided into corresponding natural segments (subsections).

14)Lines 168-170: Please add references.

Answer: Many thanks for your attention to our work. The reference [39] has been added in manuscript.

[39] Li. Z. J, He. Y, Yan. S. M, Li. H. J, Chen. J, Zhang. C, Li. C. H, Zhao. Y, Fan. Y, Guo. C. H. A novel silk fibroin-graphene oxide hybrid for reinforcing corrosion protection performance of waterborne epoxy coating, Colloids Surf. A Physicochem. Eng. Asp. 2022, 634, 127959-127974

15)Figure 2c: Did the authors detect any Raman peaks for the -F-containing groups?

Answer: Many thanks for your attention to our work. F-containing groups were detected in Raman. Because the F-containing groups peak coincides with the telescopic vibration peaks of -CH3 and -CH2 in the long-chain alkyl.

16)Do the authors know the actual percentage of modification of the GO to the other two modified structures?

Answer: Many thanks for your attention to our work. We have calculated the actual percentage of three modified structures.

According to the increased proportion, the proportions were calculated as 1.00 :1.62 :1.71.

17)For the XPS, authors should provide references for the different peaks and where are attributed.

Answer: Many thanks for your attention to our work. The article have provided references for different peaks and where they are attributed.

Fig. d1 exhibits four peaks at 284.8, 287.1, 291.3, and 293.8eV of C1s spectra [25]. Figure d2, the Si2p spectra of FGO can be divided into three peaks at 100.1, 100.9 and 102.2eV, which correspond to Si-C, Si-O, and O-Si-O bonds[41], respectively. Figure d3 shows two typical peaks at 689.4 eV and 686.9 eV, which can belong to the C-F2 and C-F3 binding energies [42].

[25] Wang S., Liu W.Q., Shi H.Y., Zhang F.Y., Liu C.H., Liang Y.Y., Pi K.. Co-modification of nano-silica and lysine on graphene oxide nanosheets to enhance the corrosion resistance of waterborne epoxy coatings in 3.5% NaCl solution, Polymer, 2021, 222, 123665-123679.

[41] An Q. F., Wang K. F., Jia Y.. Film morphology, orientation and performance of dodecyl/carboxyl functional polysiloxane on cotton substrates, Applied Surface Science, 2011, 257(63): 4569-4574.

[42] Yang B. W., Lu P., An Q. F., Zhu C.. Protective effect of silicone-modified epoxy resin coating on iron-based materials against sulfate-reducing bacteria-induced corrosion, Prog. Org. Coat, 2023, 174, 107241-107251.

18)Lines 187-188: Please add references.

Answer: Many thanks for your attention to our work. The reference has been added to the manuscript.

[25] Wang S., Liu W.Q., Shi H.Y., Zhang F.Y., Liu C.H., Liang Y.Y., Pi K.. Co-modification of nano-silica and lysine on graphene oxide nanosheets to enhance the corrosion resistance of waterborne epoxy coatings in 3.5% NaCl solution, Polymer, 2021, 222, 123665-123679.

[41] An Q. F., Wang K. F., Jia Y.. Film morphology, orientation and performance of dodecyl/carboxyl functional polysiloxane on cotton substrates, Applied Surface Science, 2011, 257(63): 4569-4574.

[42] Yang B. W., Lu P., An Q. F., Zhu C.. Protective effect of silicone-modified epoxy resin coating on iron-based materials against sulfate-reducing bacteria-induced corrosion, Prog. Org. Coat, 2023, 174, 107241-107251.

19)Lines 190-192: I am not so sure if the authors can claim that roughness has changed, since they base their explanation on round morphologies that can be seen after modification. No values are provided to sustain this statement. Moreover, what is the explanation about the differences in morphology? Authors should add something about with appropriate references.

Answer: Many thanks for your attention to our work. We have added the SEM diagram in the supporting materials Fig.S2, it can be seen that the surface of GO presents a fold-like morphology, and the silicon dioxide spherical particles with micro-nano structure are uniformly dispersed on the GO modified by silane. The morphology changes before and after GO modification, so it can be said that there is a morphological difference, and it also indicates that the GO roughness changes. References [25] in the text can be consulted.

[25] Wang S., Liu W.Q., Shi H.Y., Zhang F.Y., Liu C.H., Liang Y.Y., Pi K.. Co-modification of nano-silica and lysine on graphene oxide nanosheets to enhance the corrosion resistance of waterborne epoxy coatings in 3.5% NaCl solution, Polymer, 2021, 222, 123665-123679.

20)Lines 200-201: I do not fully understand this statement“apart from the GO nanosheet, an amorphous layer with thickness of 5~10 nm can be clearly observed around the nanosheet surface”. What do the authors want to show with this? What is this amorphous layer? Please elaborate a little bit this point.

Answer: Many thanks for your attention to our work. To demonstrate the successful modification of FGO, GO is a two-dimensional sheet crystal structure, while the layer is an amorphous silica particle modified by fluorosilane on the graphene oxide sheet.

21)Lines 206-208: This is the first point inside the text, where authors refer to the epoxy polymer. I think it is missing from the materials and methods of how the authors prepared the epoxy polymer with FGO incorporated.

Answer: Many thanks for your attention to our work. The lines 206-208 have added prepared the epoxy polymer with FGO incorporated.

First weigh 50 mg of FGO into the oven to dry and obtain FGO powder, then weigh 2.0 g of epoxy resin E51 and dilute it with ethyl acetate solvent to 30% for reserve, and then add FGO powder and epoxy curing agent D230 to 30% E51 spare resin, use a high-speed dispersing machine to disperse the mixture at high speed at 6000 rpm for 15 min. Finally, put it into the oven to heat up to 120 °C and keep it for 20 min to obtain E-FGO composite coating. .

22)Authors present a supehydrophobic contact angle of the epoxy-fgo. I think it is very important to show the contact angles of the pure epoxy, the epoxy-go and epoxy-sgo to evaluate the differences among the materials.

Answer: Many thanks for your attention to our work. The fig.5c have added the contact angles of the pure epoxy, the epoxy-go and epoxy-sgo.

Fig. 5c the effect of different proportion of (GO, SGO, FGO) on E-51 water contact angle.

The hydrophobicity increases with the addition of nano filler (GO, SGO, FGO), and it was still difficult to make the E-FGO composite layer superhydrophobic when the addition of FGO increases to 15%, which is caused by the hydrophilicity of E-51 epoxy polymer itself. In general, the FGO superhydrophobic material is well dispersed in E-51 resin, and the introduction of FGO significantly increases the hydrophobicity of the epoxy polym in Fig. 5c

23)Line 222: What is E-21? Please define before use it.

Answer: The E-51 refers to epoxy resin, which is defined in the 2.1 Material Section.

24)Figure 5c: In Figure 4a the contact angle is 151 for a 10% E-FGO and then for (as

understand same material) the contact angle is 93.5. Any comment on this?

Answer: Many thanks for your attention to our work. Figure 4a is the FGO superhydrophobic coating, and Figure 5c is a composite coating prepared by blending 10% FGO superhydrophobic coating with E51 epoxy resin because the surface of E51 coating contains hydroxyl groups and epoxy groups and other hydrophilic groups resulting in poor hydrophobic effect, but after adding FGO superhydrophobic material, the hydrophobic angle of E51 increased from 58.6° to 93.5°, and the hydrophobicity was greatly improved.

25)Line 274: Fig 7 is appearing before than its mention.

Answer: Thank you for your suggestion. Corrections have been made in the article.

26)Line 277: Same for Table 1 as for Fig 7.

Answer: Thank you for your suggestion. Revisions have been made to the article.

27)Why the 10% shows better resistance than 15%? Authors show an explanation in lines 292-298, but I think it must strengthen with references to be validated. How did they found these agglomerations?

Answer: Many thanks for your attention to our work. The introduction of FGO can considerably optimize the salt, acid, and alkaline resistance of the epoxy coating. With further increasing the E-FGO concentration to 15%, the corrosion resistance of the composite coating decreases inevitably. This is predominantly specified to the fact that FGO has a sizeable relevant surface area, and when abundant FGO is incorporated in epoxy coatings, agglomeration tends to occur. While a small amount of FGO can exert its hydrophobic effect, shielding impact, and labyrinth effect in the coating, thus significantly heightening the anti-corrosion performance of epoxy coatings. This result is consistent with the results of Figure 6a. [24] references in the text.

[24] Guo H. F., Chao B., Zhao Z. Q., Nan D.. Preparation of aniline trimer modifified graphene oxide new composite coating and study on anticorrosion performance. Mater. Res. Express, 2020, 7, 125601-125613.

Round 2

Reviewer 2 Report

I have read the revised manuscript and the responses provided by the authors. The new manuscript version has greatly improved, and the authors have responded to all my considerations. I believe that now the manuscript can be accepted.